# ATTENTIVE RECURRENT COMPARATORS

**Pranav Shyam**[*] **& Ambedkar Dukkipati**
Department of Computer Science and Automation
Indian Institute of Science
Bengaluru, India
`pranavm.cs13@rvce.edu.in, ad@csa.iisc.ernet.in`

## ABSTRACT

Attentive Recurrent Comparators (ARCs) are a novel class of neural networks built with attention and recurrence that learn to estimate the similarity of a set of objects by cycling through them and making observations. The observations made in one object are conditioned on the observations made in all the other objects. This allows ARCs to learn to focus on the salient aspects needed to ascertain similarity. Our simplistic model that does not use any convolutions performs comparably to Deep Convolutional Siamese Networks on various visual tasks. However using ARCs and convolutional feature extractors in conjunction produces a model that is significantly better than any other method and has superior generalization capabilities. On the Omniglot dataset, ARC based models achieve an error rate of 1.5% in the One-Shot classification task - a 2-3x reduction compared to the previous best models. This is also the first Deep Learning model to outperform humans (4.5%) and surpass the state of the art accuracy set by the highly specialized Hierarchical Bayesian Program Learning (HBPL) system (3.3%).

## 1 INTRODUCTION

Advancing Deep Learning systems to solve Artificial Intelligence tasks requires that models be capable of performing continual meta-learning[ (Lake et al., 2016), (Schaul & Schmidhuber, 2010)]. But top-down hierarchical designs of models (Santoro et al., 2016) to perform such tasks are not very successful on real world data and there are many reasons for this. First, most datasets are generally not designed with such higher order tasks in mind, thus researchers either work with synthetic data or fabricate higher level tasks based on traditional datasets - both of which constrain their utility. Second, hierarchical or meta models suffer from reduced supervision during training due to their inherent design. Third, with our experiments we found that the foundational architectures like Memory Augmented Neural Networks are still in their infancy and not ripe enough to be utilized in complex hierarchical systems. Therefore, in this paper, we present an alternative way of bridging this gap by building models in a bottom-up fashion. Comparing two or more inputs and estimating their similarity is a primal task using which more sophisticated models can be designed - an idea that has been well exploited in traditional Machine Learning for long (Bellet et al., 2013). Using the modern developments of attention mechanisms and by combining it with recurrent neural networks, we first built better *comparators* called Attentive Recurrent Comparators (ARCs) [1]. Using ARCs as a foundational element, we were then able to build more complex models and achieve qualitatively better results on tasks like one-shot learning. Thus, this work is proof of concept for the bottom-up design approach that can be applied to almost any dataset.

When a person is asked to compare two objects and estimate their similarity, the person does so by repeatedly looking back and forth between the two objects. With each glimpse of an object, a specific observation is made. These observations made in both objects are then cumulatively used to come to a conclusion about their similarity. A crucial characteristic of this process is that new observations are made conditioned on the previous context that has been investigated so far by the observer. The observation and it's contextual location are based on intermediate deductions. These intermediate deductions are themselves based on the observations made so far in the two objects.

---

[*]Other Affiliation: Student at R V College of Engineering, Bengaluru
[1]Code available at https://github.com/pranv/ARC

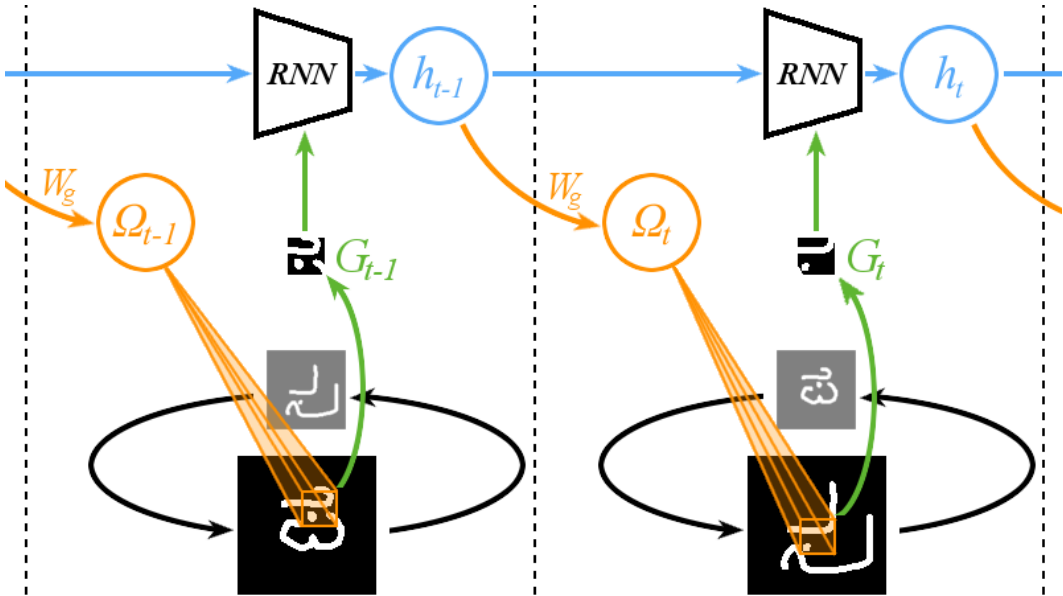

Figure 1: The abstract computational graph of a Binary ARC comparing two images. The controller which is an RNN primes the whole process. The two images are alternatively and repeatedly attended to, depicted by the carousel below. At each time-step the glimpse taken from the image is based on the attention parameters $\Omega_t$ which is calculated using the previous state of RNN $h_{t-1}$ by projecting it with $W_g$. The glimpse obtained $G_t$ and the previous state $h_{t-1}$ together used to update the state of controller to $h_t$. The vertical dotted lines demarcate the time-steps.

A series of such guided observations and the entailing inferences are accumulated and finally the judgement on similarity is made.

In stark contrast to this, current similarity estimating systems in Deep Learning are analogues of the Siamese similarity learning system (Bromley et al., 1993). In this system, a fixed set of features is detected in both the objects. Detection of features is independent of the features present in the other object. The two objects are compared based on the mutual agreement in the detected features. More concretely, comparison between two objects in this system consists of measuring the distance between their vector embeddings. A neural network defines the mapping from the object to the corresponding embedding vector in target space. This neural network is trained to extract the most salient features from the object for the specific task in hand.

There is major underlying difference between the human approach discussed above and the siamese approach to the problem. In the *human way*, the information from the two objects is fused from the very beginning and this combined information primes the subsequent steps in comparison. There are multiple lookups on each of the objects and each of these lookups are conditioned on the observations of both the objects so far. In the *siamese way*, when the embeddings in the target space are compared the information fuses mostly at an abstract level and only in the last stage.

We were interested to see the utility of the human way of comparing objects. For this, we used the modern tools of attention and recurrence to build an end-to-end differentiable model that can learn to compare objects called Attentive Recurrent Comparators (ARCs). ARCs judge similarity of objects similar to the way people do as discussed above.

We tested ARCs across many visual tasks and compared it against strong baselines of prevalent methods. ARCs which did not use any convolutions showed superior performance compared to Deep Convolutional Siamese Neural Networks on challenging tasks. Though Dense ARCs are as capable as ConvNets, a combination of both ARCs and convolutions produces more superior models (hereafter referred to as ConvARCs), capable of better generalization and performance. In the task of estimating the similarity of two characters from the Omniglot dataset for example, ARCs and Deep Convnets both achieve about 93.4% accuracy, whereas ConvARCs achieve 96.10% accuracy.

Further, as discussed above, similarity estimation is a generic and a primal task in many other higher-level cognitive tasks. Evaluating our model on these higher-level tasks also lets us explore the generalisation capacity of ARCs. In this work, we study the performance of models designed to perform One Shot Learning with ARCs as building blocks. On the Omniglot one-shot classification task, our model achieved 98.5% accuracy significantly surpassing the current state of the art set by Deep Learning methods or other systems.

Fundamentally, the performance of ARCs shows the value of early fusion of information across the entire context of the task. Further, it also strengthens the view that attention and recurrence together can be as good as convolutions in some cases.

## 2 ATTENTIVE RECURRENT COMPARATORS

The ARC model can be directly derived by distilling the vital aspects from the human way discussed in Section 1. In the following paragraphs we describe the ARC model for the binary image case - where there are two images whose similarity has to be judged. It is trivial to generalise it to more objects or other modalities. See Figure 1 for a visual depiction of the model.

The model operates on given two images over the span of an *episode*. The images are given at the beginning of the episode and the ARC is expected to emit a token of similarity at the end of this episode. Given two images $\{x_a, x_b\}$, the model repeatedly cycles through the both, attending to only one image at one time step. Thus the sequence of presentations is $x_a, x_b, x_a, x_b, ...$ and so on for a finite number of presentations of each image. An episode is nothing more than a collection of time-steps, with an action being taken in each time-step.

For time step $t$ the input image presented is given by:

$$I_t \longleftarrow x_a \text{ if } t \% 2 \text{ is } 0 \text{ else } x_b$$

The model functionally consists of a recurrent core and an attention mechanism. During the span of the episode, the model iteratively focuses its attention on the current input. At each time step of the episode, the model attends to only one input, but over the course of many time steps it would have observed many aspects of all the inputs. The observations are made by the model at each time step by directing its attention to a region of interest in each input. Since the core of the model is a recurrent neural network, this round robin like cyclic presentation of inputs allows for early fusion of information from all the inputs. This makes the model aware of the context in which it is operating. Consequently, this provides feedback to the attention mechanism to attend on the relevant and crucial parts of each sample considering the context of all the inputs and observations made so far.

If there are $n$ inputs and we allow for $g$ glimpses of each input, then the episode length $L$ is $ng$. The hidden state of the RNN controller at the final time step $h_L$ can be then used for subsequent processing.

The attention mechanism focuses on a specific region of the image $I_t$ to get the glimpse $G_t$.

$$G_t \longleftarrow attend(I_t, \Omega_t) \qquad \Omega_t = W_g h_{t-1}$$

$attend(.)$ is the attention mechanism described in the sub section below, that acts on image $I_t$. $\Omega_t$ are the attention glimpse parameters which specify the location and size of the attention window. At each step, we use the previous hidden state of the RNN core $h_{t-1}$ to compute $\Omega_t$. $W_g$ is the projection matrix that maps the hidden state to the required number of attention parameters.

Next, both the glimpse and previous hidden state are combined to form the next hidden state.

$$h_t \longleftarrow RNN(G_t, h_{t-1})$$

$RNN(.)$ is the update function for the recurrent core being used. This could be simple RNN or an LSTM.

The above 4 equations describe the Binary ARC. We arrived at the iterative cycling of input paradigm after trying out many approaches to attend to multiple images at once. Iterative cycling turned out to

be more computationally efficient, scalable and statistically more consistent than other approaches we tested. Note that $I_t$ for some $t$ alternates between $x_a$ and $x_b$, while the rest of the equations are exactly the same for all time steps.

## 2.1 Attention Mechanism

The attention mechanism is based on DRAW (Gregor et al., 2015), zoomable and differentiable. The attention window is defined by an $N \times N$ 2D grid of Cauchy kernels. We found that the heavy tail of the Cauchy curve to aids in alleviating some of the vanishing gradient issues and it sped up training.

The grid's location and size is defined based on the glimpse parameters. The $N \times N$ grid of kernels is placed at $(x, y)$ on the $S \times S$ image, with the central Cauchy kernel being located at $(x, y)$. The distance between two Cauchy kernals either in the vertical or horizontal direction is $\delta$. In other words, the elemental square of the 2D grid is $\delta \times \delta$ in size. The glimpse parameter set $\Omega_t$ is unpacked to get $\Omega_t \rightarrow (\widehat{x}, \widehat{y}, \widehat{\delta})$. $x, y$ and $\delta$ are computed from $\widehat{x}, \widehat{y}$ and $\widehat{\delta}$ using the following transforms:

$$x = (S-1)\frac{(\widehat{x}+1)}{2} \qquad y = (S-1)\frac{(\widehat{y}+1)}{2} \qquad \delta = \frac{S}{N}(1-|\widehat{\delta}|) \qquad \gamma = e^{1-2|\widehat{\delta}|}$$

The location of a $i^{th}$ row, $j^{th}$ column's Cauchy kernel in terms of the pixel coordinates of the image is given by:

$$\mu_X^i = x + (i - (N+1)/2)\,\delta \qquad \text{and} \qquad \mu_Y^j = y + (j - (N+1)/2)\,\delta$$

The horizontal and vertical filterbank matrices are then calculated as:

$$F_X[i,a] = \frac{1}{Z_X}\left\{\pi\gamma\left[1 + \left(\frac{a-\mu_X^i}{\gamma}\right)^2\right]\right\}^{-1} \text{ and } F_Y[j,b] = \frac{1}{Z_Y}\left\{\pi\gamma\left[1 + \left(\frac{b-\mu_Y^j}{\gamma}\right)^2\right]\right\}^{-1}$$

$Z_X$ and $Z_Y$ are normalization constants such that they make $\Sigma_a F_X[i,a] = 1$ and $\Sigma_b F_X[j,b] = 1$

Final result of the attending to the image is given by:

$$attend(I_t, \Omega_t) = F_Y I_t F_X^T$$

*attend* thus gets an $N \times N$ patch of the image, which is flattened and used in the model.

## 2.2 Use of Convolutions

As seen in the experimental section below, while simple attention over raw images performs as well as Deep ResNets, we found large improvements by using Convolutional feature extractors. Applying several layers of Convolution produces a 3D solid of activations (or a stack of 2D feature maps). Attention over this corresponds to applying the same 2D attention over the entire depth of the 3D feature map and outputting the flattened glimpse.

## 3 Practical Experiments and Analysis

Understanding the empirical functioning of an ARC and identifying factors affecting its performance requires both qualitative and quantitative studies. Qualitative analysis tells us what the model is doing when it is comparing 2 images and how this relates to human ways of comparison. Quantitative analysis shows the variations in performance when certain aspects of the model are changed and thus provide an estimate of their importance. For the analysis presented below, we use the simple ARC model (without convolutions) described in Section 2 above trained for the verification task on the Omniglot dataset. Data samples in the Omniglot dataset have an understandable structure with characters being composed of simple strokes drawn on a clean canvas. The dataset is also very diverse, which allows us to study various characteristics of our model under a wide range of conditions. Since our main result in the paper is also on the Omniglot dataset (Sections 4 and 5), we train

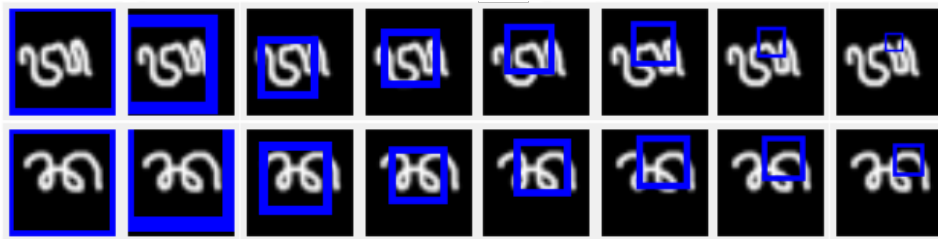

(a) It can be seen that the two characters look very similar in their stroke pattern and differ only in their looping structure. ARC has learnt to focus on these crucial aspects.

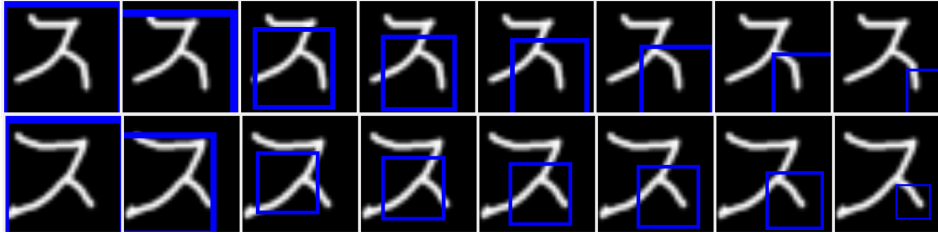

(b) ARC parses over the characters in a left to right, top to bottom fashion. Finally, it ends up focussing in the region where the first character has a prolonged downward stroke, whereas the second one does not.

Figure 2: Attention windows over time when comparing the two Omniglot characters. The top row has the first image and the bottom row has the second. Each column represents a glimpse step. (a) Comparing two dissimilar characters and (b) Comparing two similar characters.

the ARC on this same dataset for this analysis to get an insight into the type of performance gains brought about by this architecture.

The verification task is a binary classification problem wherein the model is trained to predict whether the 2 drawings of characters provided belong to the same character or not (see Section 4 for more details). The final hidden state of the RNN Controller $h_L$ is given to a single logistic neuron to estimate the probability of similarity. The whole setup is trained end to end with back-propagation and SGD. The particular model under consideration had an LSTM controller (Hochreiter & Schmidhuber, 1997) with forget gates (Gers et al., 2000). The number of glimpses per image was fixed to 8, thus the total number of recurrent steps being 16. $32 \times 32$ greyscale images of characters were used and the attention glimpse resolution is $4 \times 4$.

### 3.1 QUALITATIVE ANALYSIS

The following inferences were made after studying several cases of ARC's operation (see Figure 2 for an example):

1. The observations in one image are definitely being conditioned on the observations in the other image. This can be seen in figures 2a and 2b.

2. The ARC seems to have learnt a fairly regular *left to right* parsing strategy, during which the attention window gradually reduces in size. This is quite similar to strategies found in other sequential attentive models like DRAW (Gregor et al., 2015).

3. Deviation from such regular ordered parsing occurs if model finds some interesting feature in either character. This results in attention being fixated to that particular region of the character for a few subsequent glimpses.

4. There is no strict coordination or correspondence chronologically between the attended regions of the two images. While instances of ARC focussing on the same aspect/stroke of two characters were common, there were plenty more instances wherein the ARC attended to different aspects/strokes in each image during an interval. We hypothesise that the RNN

controller could be utilizing turns of glimpsing at an image to observe some other aspects which are not of immediate consequence.

5. We also frequently encountered cases wherein the attention window, after parsing as described in point 2, would end up focusing on some blank, stroke-less region, as if it had stopped looking at the sample. We hypothesize that the model is preferring to utilize its recurrent transitions and not to be disturbed by any input stimuli.

## 3.2 QUANTITATIVE ANALYSIS

We performed a simple yet very insightful ablation study to understand ARC's dynamics. ARC accumulates information about both the input images by a series of attentive observations. We trained 8 separate binary classifiers to classify images as being similar or not based on hidden states of the LSTM controller at every even time step correspondingly . The performance of these binary classifiers are correlated with the information contained in the hidden states. The performance of these classifiers is reported in Table 1. Since the ARC has an attention window of only $4 \times 4$ pixels, it can barely see anything in the first time step, where its attention is spread throughout the whole image. With more glimpses, finer observations bring in more precise information into the ARC and the recurrent transitions make use of this knowledge, leading to higher accuracies. We also used the 8 binary classifiers to study how models confidence grows with more glimpses and one such good example is provided in Figure 3.

Table 1: Glimpses per image vs Classification Accuracy of ARC

| GLIMPSES | ACCURACY |
|:---:|:---:|
| 1 | 58.2% |
| 2 | 65.0% |
| 4 | 80.8% |
| 6 | 89.25% |
| **8** | **92.08%** |

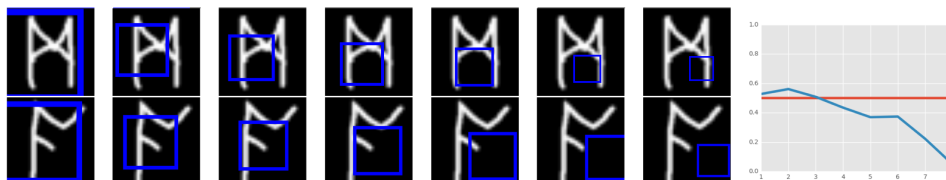

(a) ARC is very unsure of similarity at the beginning. But at 5th glimpse (4th column), the attention goes over the region where there are strokes in the first image and no strokes in the second one resulting in dropping of the score.

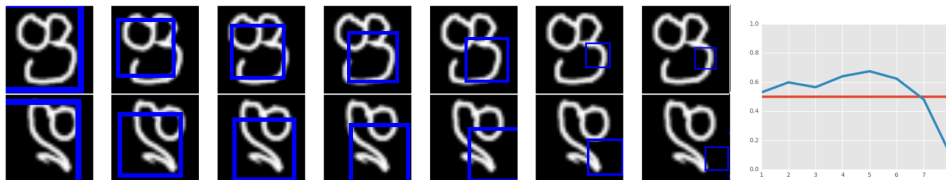

(b) Initially ARC is unsure or thinks that the characters are similar. But towards the end, at 6th glimpse (5th column), the model focusses on the region where the connecting strokes are different. The similarity score drops and with more "ponder", it falls down significantly.

Figure 3: Attention windows over time and instantaneous predictions from independent binary classifiers. The first glimpse is omitted as it covers the whole image. In the graph: x-axis - glimpse number, y-axis - similarity score. The red line is the decision threshold, above which the images are considered to be similar. Both of the cases above are examples of a dissimilar pair.

## 4 SIMILARITY LEARNING

Verification is a generic and common task in Machine Learning. The verification task essentially requires models that can predict whether the two inputs are the same or different, for some notion of same (such as unique facial identity, objects of same class etc.,). Specifically, here we restrict ourselves to the task of estimating the similarity of the given pair of images. When given two images the models are required to output a single logistic value, which is expected to be 1 for very similar inputs and 0 for very dissimilar inputs. We compare our ARC model with several baselines and report performance on two challenging datasets.

### 4.1 DATASETS

#### 4.1.1 OMNIGLOT

The dataset is thoroughly detailed in the next section which is on one shot classification on this dataset. And this task acts as a precursor to the more sophisticated next task. We use $32 \times 32$ images and similar/dissimilar pairs of character drawings are randomly chosen only within alphabet to make the task more challenging. Out of the 50 alphabets provided in the dataset, 30 were used for training, 10 for validation and the last 10 for testing.

### 4.2 BASELINES AND IMPLEMENTATION DETAILS

We consider strong convolutional baselines, which have been shown time and again to excel at such visual tasks. We particularly use Wide Resnets (WRNs) (Zagoruyko & Komodakis, 2016) which are the current state of the art models in image classification. Independent nets were tuned for each dataset. Hyper parameters were set for reasonable values for all our ARC models and no hyperparameter tuning of any kind was employed. For the Omniglot dataset, we also include the result from (Koch et al.) We used moderate data augmentation consisting of translation, flipping, rotation and shearing, which we found to be critical for training ARC models.

### 4.3 RESULTS

The results are in Table 1 for Omniglot respectively. Our simple ARC model without using any convolutional layers obtains a performance that matches a AlexNet style 6 layer Deep Convnet with millions of parameters. Using convolutional feature extractors, ARCs outperform the Wide ResNet based Siamese ConvNet baselines, even the ones containing an order of magnitude more parameters.

Table 2: Performance of ARC vs conventional methods on the verification task. All values are accuracies on the test set. For Wide ResNets, suffixes specify the depth and width. For example, *(d=60, w=4)* means that it is a ResNet that 60 is layers deep with each residual block having a width multiplier of 4.

(a) Omniglot Dataset

| MODEL | ACCURACY |
|---|---|
| Siamese Network | 60.52% |
| Deep Siamese Net (Koch et al.) | 93.42% |
| Siamese ResNet (d=24, w=1) | 93.47% |
| Siamese ResNet (d=30, w=2) | 94.61% |
| Siamese ResNet (d=60, w=4) | 93.57% |
| **ARC** | **93.31%** |
| **ConvARC** | **96.10%** |

## 5 OMNIGLOT ONE SHOT CLASSIFICATION

One shot learning requires Machine Learning models to be at the apotheosis of data efficiency. In case of classification, only a single example of each individual class is given and the model is

expected to generalise to new samples. A classic example is of a human kid learning about the animal giraffe (Vinyals et al., 2016). The kid does not need to see thousands of images of a Giraffe to learn to detect it. Rather, just from a single example, the kid can not only recognize it at a future point, but going further, she can also speculate on its other characteristics. While humans excel at this task, current Deep Learning systems are at the opposite end of the spectrum, where they are trained on millions of samples to achieve the kind of results that they are well known for. With ARCs we have developed a generic method to comparing objects. We have also shown that our model generalizes extremely well. So we decided to test ARC on the challenging Omniglot dataset.

Omniglot is a dataset by Lake et al. (2015) that specially designed to compare and contrast the learning abilities of humans and machines. The dataset contains handwritten characters of 50 of the world's languages/alphabets. Though there are 1623 characters, there are only 20 samples for each which is drawn by 20 individuals. So this is in a diagonally opposite position when compared to MNIST or ImageNet. One Shot Classification on this dataset is very challenging one as most Deep Learning systems do not work well in such extreme conditions. Lake et al. (2015) developed a dedicated system for such rapid knowledge acquisition called Hierarchical Bayesian Programming Learning, which surpasses human performance and is the current state of the art of all methods.

## 5.1 TASK

The dataset is divided into a background set and an evaluation set. Background set contains 30 alphabets (964 characters) and only this set should be used to perform all learning (e.g. hyper-parameter inference or feature learning). The remaining 20 alphabets are for pure evaluation purposes only. Each character is a $105 \times 105$ image.

A one shot classification task episode is as follows: from a randomly chosen alphabet, 20 characters are chosen which becomes the support set. One character among these 20 becomes the test character. 2 drawers are chosen, one each for the support set and the test character. The task is to match the test drawing to the correct character's drawing in the support set. Assigning an image to one of the 20 characters given results in a 20-way, 1-shot classification task.

## 5.2 MODELS

### 5.2.1 NAIVE ARC MODEL

This is a trivial extension of ARC for used for verification to this task. The test image from the first set is chosen and compared against all 20 images from the second set. It is matched to the character with maximum similarity. This is done for 20 times for each character in the first set.

### 5.2.2 FULL CONTEXT ARC

Our whole hypothesis in this work has been about the value of providing the full context to the model. And we have shown to that models which are aware of the context of operation are better than the ones that aren't. While Naive ARC model is simple and efficient, it does not incorporate the whole context in which our model is expected to make the decision of similarity. When the character is being compared to 20 other characters from the support set, the comparisons are all independently done. That is, the model is not aware available options for matching, so it assigns the similarity score to each pair independently.

It is highly desirable to have a 20-way ARC, where each observation is conditioned on the all images. Unfortunately, such a model is not practical. The recurrent controller has memory limitations in its state. Scaling up the memory incurs a huge parameter burden. So instead, we use a hierarchical setup, which decomposes the comparisons to be at two levels - first local pairwise comparison and a second global comparison. We found that this model reduces the information that has to be crammed in the controller state, while still providing sufficient context.

As with the Naive method, we compare one image from set A with one from set B in pairs. But instead of emitting a similarity score immediately, we collect the *comparison embeddings* of each comparison. Comparison embeddings are the final hidden state of the controller when the test image $T$ is being compared to image $S_j$ from the support set B: $e_j = {h_L}^{T,S_j}$. These embeddings are further processed by a Bi-Directional LSTM layer. This merges the information from all comparisons, thus

providing the necessary context before score emission. This is also the method used in Matching Networks (Vinyals et al., 2016).

$$c_j = [\,\overrightarrow{LSTM}(e_j);\ \overleftarrow{LSTM}(e_j)\,] \qquad \forall j \in [1, 20]$$

Each embedding is mapped to a single score $s_j = f(c_j)$, where $f(.)$ is an affine transform followed by a non-linearity. The final output is the normalized similarity with respect to all similarity scores.

$$p_j = softmax(s_j) \qquad \forall j \in [1, 20]$$

This whole process is to make sure that we adhere to the fundamental principle of deep learning, which is to optimise objectives that directly reflect the task. The normalisation allows for the expression of relative similarity rather than absolute similarity.

## 5.3 BASELINES AND OTHER METHODS

We compare the two models discussed above with other methods in literature: starting from the simplest baseline of k-Nearest Neighbours to the latest meta-learning methods. The training and evaluation practises are non consistent.

### 5.3.1 ACROSS ALPHABETS

Many papers recently, like Matching Networks Vinyals et al. (2016) and MANNs Santoro et al. (2016) have used 1200 chars for background set (instead of 964 specified by Lake et al. (2015)). The remaining 423 characters are used for testing. Most importantly, the characters sampled for both training and evaluation are across all the alphabets in the training set.

### 5.3.2 WITHIN ALPHABETS

This corresponds to standard Omniglot setting where characters are sampled within an alphabet and only the 30 background characters are used for training and validation.

The across alphabet task is much more simpler as it is easy to distinguish characters belonging to different languages, compared to distinguishing characters belonging to the same language. Further, across alphabet methods use a lot more data which is a particularly advantageous entity for Deep Learning Methods.

There are large variations in the resolution of the images used as well. The Deep Siamese Network of Koch et al. uses 105x105 images and thus not comparable to out model, but we include it as it is the current best result using deep neural nets. The performance of MANNs in this standard setup is interpreted from the graph in the paper as the authors did not report it. It should also be noted that HBPL incorporates human stroke data into the model. Lake et al estimate the human performance to be at 95.5%.

Table 3: One Shot Classification accuracies of various methods and our ARC models.

(a) Across Alphabets

| MODEL | ACCURACY |
|---|---|
| kNN | 26.7% |
| Conv Siamese Network | 88.1% |
| MANN | $\approx 60\%$ |
| Matching Networks | 93.8% |
| Naive ARC | 90.30% |
| **Naive ConvARC** | **96.21%** |
| **Full Context ConvARC** | **97.5%** |

(b) Within Alphabet

| MODEL | ACCURACY |
|---|---|
| kNN | 21.7% |
| Siamese Network | 58.3% |
| Deep Siamese Network (Koch et al.) | 92.0% |
| Humans | 95.5% |
| HBPL | 96.7% |
| Naive ARC | 91.75% |
| **Naive ConvARC** | **97.75%** |
| **Full Context ConvARC** | **98.5%** |

## 5.4 RESULTS

Results are presented in Table 2. Our ARC models outperform all previous methods according to both of the testing protocols and establish the corresponding state of the art results.

## 6 RELATED WORK

Deep Neural Networks (Schmidhuber, 2015) (LeCun et al., 2015) are very complex parametrised functions which can be adapted to have the required behaviour by specifying a suitable objective function. Our overall model is a simple combination of the attention mechanism and recurrent neural networks (RNNs). We test our model by analysing its performance in similarity learning. We also test its generalisation ability by using it in a model built for the challenging task of one shot classification on hand-written character symbols.

### 6.1 ATTENTION

It is known that attention brings in selectivity in processing information while reducing the processing load (Desimone & Duncan, 1995). Attention and (Recurrent) Neural Networks were combined in Schmidhuber & Huber (1991) to learn fovea trajectories. Later attention was used in conjunction with RBMs to learn what and where to attend in Larochelle & Hinton (2010) and in Denil et al. (2012). Hard Attention mechanism based on Reinforcement Learning was used in Mnih et al. (2014) and further extended to multiple objects in Ba et al. (2014); both of these models showed that the computation required at inference is significantly less compared to highly parallel Convolutional Networks, while still achieving good performance. A soft or differentiable attention mechanisms have been used in Graves (2013). A specialised form of location based soft attention mechanism, well suited for 2D images was developed for the DRAW architecture (Gregor et al., 2015), and this forms the basis of our attention mechanism in ARC.

### 6.2 SIMILARITY LEARNING

A survey of the methods and importance of measuring similarity of samples in Machine Learning is presented in Bellet et al. (2013). With respect to deep learning methods, the most popular architecture family is that of Siamese Networks (Bromley et al., 1993). The energy based derivation is presented in Chopra et al. (2005) and since then they have been used across wide range of modalities - in vision (Zagoruyko & Komodakis, 2015) (Bertinetto et al., 2016), for face recognition and verification (Taigman et al., 2014) and in Natural Language Processing (Lu & Li, 2013) (Hu et al., 2014). Recently Triplet Losses (Hoffer & Ailon, 2015) are being used to achieve higher performance and is similar to our Ternary ARC model at an abstract level.

### 6.3 ONE SHOT LEARNING

A bayesian framework for one shot visual recognition was presented in Fe-Fei et al. (2003). Lake et al. (2015) extensively study One Shot Learning and present a novel probabilistic framework called Hierarchical Bayesian Program Learning (HBPL) for rapid learning. They have also released the Omniglot dataset, which has become a testing ground for One Shot learning techniques. Recently, many Deep Learning methods have been developed to do one shot learning: Koch et al. use Deep Convolutional Siamese Networks for performing one shot classification. Matching Networks (Vinyals et al., 2016) and Memory Augmented Neural Networks (Santoro et al., 2016) are other approaches to perform continual or meta learning in the low data regime. All the models except the HBPL have inferior one shot classification performance compared to humans on the Omniglot Dataset.

## 7 CONCLUSION AND FUTURE WORK

We presented a model that uses attention and recurrence to cycle through a set images repeatedly and estimate their similarity. We showed that this model is not only viable but also much better than the siamese neural networks in wide use today in terms of performance and generalization. Our main

result is in the task of One Shot classification on the Omniglot dataset, where we achieved state of the art performance surpassing HBPL's and human performance.

One potential downside of this model is that due to sequential execution of the recurrent core and by the very design of the model, it might be more computationally expensive than a distance metric method. But we believe that advancing hardware speeds, such costs will be outweighed by the benefits of ARCs.

Though presented in the context of images, ARCs can be used in any modality. There are innumerable ways to extend ARCs. Better attention mechanisms, higher resolution images, different datasets, hyper-parameter tuning, more complicated controllers etc are the simple things which could be employed to achieve better performance.

More interesting extensions would involve developing more complex architectures using this bottom-up approach to solve even more challenging AI tasks.

ACKNOWLEDGEMENTS

We would like to thank all the members of the Statistics and Machine Learning Lab at the Indian Institute of Science for their support and feedback. We would like to specifically thank Akshay Mehrotra for his extensive help with everything from the implementation to discussing results. We would also like to thank Siddharth Agrawal and Gaurav Pandey for their helpful feedback throughout the process. We would like to thank Soumith Chintala for his feedback on this manuscript and the idea.

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
