# Peer review of "Attentive Recurrent Comparators"

_ICLR 2017 — rejected_

[Official Review · AnonReviewer3 · rating 3 · confidence 5 · 14 Dec 2016]
**The paper need more improvements to be accepted**

This paper describes a method that estimates the similarity between a set of images by alternatively attend each image with a recurrent manner. The idea of the paper is interesting, which mimic the human's behavior. However, there are several cons of the paper:

1. The paper is now well written. There are too many 'TODO', 'CITE' in the final version of the paper, which indicates that the paper is submitted in a rush or the authors did not take much care about the paper. I think the paper is not suitable to be published with the current version.

2. The missing of the experimental results. The paper mentioned the LFW dataset. However, the paper did not provide the results on LFW dataset. (At least I did not find it in the version of Dec. 13th)

3. The experiments of Omniglot dataset are not sufficient. I suggest that the paper provides some illustrations about how the model the attend two images (e.g. the trajectory of attend).

[Reviewer Comment · AnonReviewer3 · 14 Dec 2016]
**what is the results on LFW dataset?**

It is weird that the paper describe LFW dataset but do not provides the results on it.

[Author Response · Pranav Shyam · 15 Dec 2016]
**Updated Paper**

An updated version of the paper taking into consideration the reviewers comments has been uploaded

[Official Review · AnonReviewer1 · rating 5 · confidence 2 · 16 Dec 2016]
**Strong experimental results, but somewhat unclear where the improvements are coming from**

This paper presents an attention based recurrent approach to one-shot learning. It reports quite strong experimental results (surpassing human performance/HBPL) on the Omniglot dataset, which is somewhat surprising because it seems to make use of very standard neural network machinery. The authors also note that other have helped verify the results (did Soumith Chintala reproduce the results?) and do provide source code.

After reading this paper, I'm left a little perplexed as to where the big performance improvements are coming from as it seems to share a lot of the same components of previous work. If the author's could report result from a broader suite of experiments like in previous work (e.g matching networks), it would much more convincing. An ablation study would also help with understanding why this model does so well.

[Official Review · AnonReviewer2 · rating 4 · confidence 5 · 17 Dec 2016]
**experimental section improved but still very weak on analysis and insight**

This paper introduces an attention-based recurrent network that learns to compare images by attending iteratively back and forth between a pair of images. Experiments show state-of-the-art results on Omniglot, though a large part of the performance gain comes from when extracted convolutional features are used as input.

The paper is significantly improved from the original submission and reflects changes based on pre-review questions. However, while there was an attempt made to include more qualitative results e.g. Fig. 2, it is still relatively weak and could benefit from more examples and analysis. Also, why is the attention in Fig. 2 always attending over the full character?  Although it is zooming in, shouldn’t it attend to relevant parts of the character?  Attending to the full character on a solid background seems a trivial solution where it is then unclear where the large performance gains are coming from.

While the paper is much more polished now, it is still lacking in details in some respects, e.g. details of the convolutional feature extractor used that gives large performance gain.

[Author Response · Pranav Shyam · 08 Jan 2017 (modified: 26 Feb 2017)]
**Added Analysis**

We have added a 2+ page detailed analysis section with ablation studies and attention maps.

[Final Decision · Program Chairs · 06 Feb 2017]
**ICLR committee final decision**

This paper shows some strong performance numbers, but I agree with the reviewers that it requires more analysis of where those gains come from. The model is very simple, which is a positive, but more studies such as ablation studies and other examples would help a lot.